# *Haplosporidium pinnae* Parasite Detection in Seawater Samples

**DOI:** 10.3390/microorganisms11051146

**Published:** 2023-04-28

**Authors:** Irene Moro-Martínez, Maite Vázquez-Luis, José Rafael García-March, Patricia Prado, Milena Mičić, Gaetano Catanese

**Affiliations:** 1LIMIA-IRFAP Laboratorio de Investigaciones Marinas y Acuicultura—Govern de les Illes Balears, 07157 Port d’Andratx, Spain; imoro@sgaip.caib.es; 2IEO-CSIC, Centro Oceanográfico de Baleares Instituto Español de Oceanografía, 07010 Palma de Mallorca, Spain; maite.vazquez@ieo.csic.es; 3IMEDMAR-UCV Instituto de Investigación en Medio Ambiente y Ciencia Marina, Universidad Católica de Valencia, 03710 Calpe, Spain; jr.garcia@ucv.es; 4IRTA-Sant Carles de la Ràpita, 43540 La Ràpita, Spain; patricia.prado@irta.cat; 5Aquarium Pula d.o.o., Ulica Verudella 33, 52100 Pula, Croatia; milena.micic@aquarium.hr; 6INAGEA (UIB)-Instituto de Investigaciones Agroambientales y de Economía del Agua, Universidad de las Islas Baleares, Carretera de Valldemossa, km 7.5, 07122 Palma, Spain

**Keywords:** *Haplosporidium pinnae*, eDNA, ribosomal unit, *Pinna nobilis*, benthic invertebrate, critically endangered

## Abstract

In this study, we investigated the presence of the parasite *Haplosporidium pinnae*, which is a pathogen for the bivalve *Pinna nobilis*, in water samples from different environments. Fifteen mantle samples of *P. nobilis* infected by *H. pinnae* were used to characterize the ribosomal unit of this parasite. The obtained sequences were employed to develop a method for eDNA detection of *H. pinnae*. We collected 56 water samples (from aquaria, open sea and sanctuaries) for testing the methodology. In this work, we developed three different PCRs generating amplicons of different lengths to determine the level of degradation of the DNA, since the status of *H. pinnae* in water and, therefore, its infectious capacity are unknown. The results showed the ability of the method to detect *H. pinnae* in sea waters from different areas persistent in the environment but with different degrees of DNA fragmentation. This developed method offers a new tool for preventive analysis for monitoring areas and to better understand the life cycle and the spread of this parasite.

## 1. Introduction

In the past few years, the newly described haplosporidan parasite species *Haplosporidium pinnae* were likely associated with the fan mussel *Pinna nobilis* massive mortality event (MME) [1]. Commonly, the Haplosporidia have been considered responsible for mass mortality events globally due to their pathogenic nature, parasitizing marine and freshwater invertebrates, such as *Haplosporidium nelsoni* for *Crassotrea virginica* on the east coast of the USA or *Bonamia ostreae* and *Bonamia exitosa*, reported to have infected various oyster species [2,3].

The first observations of *P. nobilis* MME were reported in September–October 2016 in the south and south-east of the Iberian Peninsula (Andalusia, Region of Murcia and southern Valencia Community) and the Balearic Islands (Formentera and Ibiza) [4,5]. Later, MME spread rapidly eastwards, affecting in 2017 the populations of Spain, France, Italy and Tunisia [1,6,7,8,9]. In 2018, a similar situation was reported in Malta, Greece, Cyprus and Turkey, and finally in 2019, the first cases of massive mortality were observed in the Adriatic Sea [10,11]. Nowadays, many of the populations of *P. nobilis* in the Mediterranean Sea have decreased by 80–100% [5,7,8,10,12].

Within a short time, span of 18 months (from 2016/17 to 2018), the MME was reported to spread from the western to the eastern Mediterranean populations of fan mussels [10].

As a result, the species *P. nobilis* was changed from the “Vulnerable” category to “Critically Endangered” in the Spanish legislation, with a serious extinction risk (Orden TEC/1078/2018), and was included as a “Critically Endangered” species on the IUCN Red List in 2019 [10].

In the past few years, some authors have reported infection and coinfection of *Mycobacterium* sp. or other pathogens, associated with the mortality episodes of *P. nobilis* [6,12,13,14,15,16]. However, recent new research pointed out that the onset of the MME in *P. nobilis* is strongly associated with the presence of *H. pinnae*, which exhibits a preeminent role compared to the other studied pathological agents [17,18,19]. *H. pinnae* apparently induces the death of *P. nobilis*, obstructing its digestive gland and provoking an inflammatory response and general dysfunction [1]. The ecological role of *P. nobilis* is paramount as it filters water and retains large amounts of detritus and a high percentage of organic matter, contributing to water transparency [20,21]. Furthermore, *P. nobilis* has been observed to have a role as a potential “island” and “source” of biodiversity of epizoan mollusks, often present in early and structured assemblages [22,23]. In general, *P. nobilis* is a filter feeder and it seems to ingest detritus, phytoplankton, micro- and meso-zooplankton and pollen grains, with varying prevalence, depending on the area and the season [24,25].

A possible life cycle stage for *H. pinnae* was proposed by Grau et al. (2022) [18], based on previous knowledge of the life cycle of similar haplosporidans [26]. Uninucleate and binucleate stages of *H. pinnae* were observed throughout the connective tissue and hemolymph of *P. nobilis*, with spherical to elongated shapes and a maximum length of 3.5 µm. Sporulation stages, including sporonts, sporocysts and spores, as well as pre-sporulation, uninucleate cells, binucleate and multinucleate plasmodia were observed inside the digestive gland tubules of *P. nobilis* [1]. Spores have been observed to range between 3.6 and 5.7 µm length and between 2.7 and 4.5 µm width and to be apparently released in the lumen of the gland’s tubules and eliminated to the environment through the intestine in fecal excretions [1,7].

However, some aspects of this parasite still remain unknown in relation to the origin, transmission and life stages as the presence in intermediate hosts, or the persistence of the spores or cells in the environment. Therefore, approaches to detect and monitor the parasite in seawater are still lacking today.

A molecular method for detection through PCR assays of *H. pinnae* DNA from *P. nobilis* tissues was already developed [1,7,27]. For its detection, the molecular marker used is the ribosomal 18S gene, which is the structural RNA for the small component of eukaryotic cytoplasmic ribosomes. Ribosomes are cell organelles made up of different types of RNA and proteins in which protein synthesis takes place, taking part in the translation of messenger RNAs (mRNAs) transcribed from genes. The ribosomal 18S (SSU), 5.8S and 28S (LSU) genes, very conserved at the species level, are arranged one after the other, separated by more variable spacer sequences (ITS1 and ITS2). This DNA region forms what is called the “tandemly repeated ribosomal unit” and it is flanked by two external spacer regions (5′ETS and 3′ETS), which separate more ribosomal units. The rDNA region allows us to carry out studies for phylogenetic and phylogeographic analysis of populations in invertebrate organisms [28,29,30], primates [31], plants and algae [32,33].

In general, in marine organisms, the 18S rDNA is mainly used for taxonomic studies, while ITS regions are used mainly for diversity studies and as barcode markers. However, compared to 18S, ITS is more variable and therefore more suitable for measuring intraspecific genetic diversity [34,35].

Developing an environmental DNA (eDNA)-based method seems to be the one and only approach to use for detecting *H. pinnae* in water. The eDNA refers to the DNA that can be extracted from environmental samples such as soil, water, air, etc., without the need to first isolate a target organism. It is a complex moisture of genomic DNA (cellular or extracellular) [36]. This methodology allows rapid, non-invasive and cost-efficient detection and monitoring of some species in seawater [37]. Some studies of eDNA focused on seawater have already revealed new parasite clades, parasite diversity and pathogen survival outside the host and have been used to investigate the life cycle of non-cultivable micro-parasites [38,39,40,41]. Nevertheless, using eDNA to detect the presence or abundance of target organisms requires knowledge of DNA decomposition patterns in aquatic environments causing a reduction in the detectable amount of eDNA and physical changes within the molecule [42].

Although a method starting from eDNA samples for identifying the congeneric species *P. nobilis* and *P. rudis* has been developed [43], no studies or methodologies are currently available to determine the presence of *H. pinnae* in water samples. Likewise, its preservation status based on the integrity of its detected DNA is unknown. Therefore, it seems useful to apply different regions of the ribosomal DNA unit as markers in eDNA analysis for detecting *H. pinnae* in water, also considering that due to its conserved region characteristic, the amplification of different DNA fragments will guarantee the identification of the species but also the detection of any intraspecific variables. Thus, obtaining amplifications of different sizes could be a valid index of the degradation status of the parasite in water.

This work aimed to characterize the whole ribosome unit of *H. pinnae* and the application of different PCRs to water samples, generating different sizes of amplicons that allow its detection (in any of its forms or state of degradation), from different environments. The developed methodology will help to improve the management/restoration of *P. nobilis* by allowing the early detection of the parasite within the sanctuaries and identifying areas free of the parasite where the translocation of juvenile individuals from collectors or future culture would be possible. The results will also help the scientific community and the administrations responsible for conservation in the choice of potential areas for reintroducing the species.

## 2. Materials and Methods

### 2.1. Sampling

For this study, fifteen mantle samples of *P. nobilis* infected by *H. pinnae*, representative of different Mediterranean open sea areas, were collected between June 2017 and November 2021. Furthermore, samples for seawater eDNA molecular diagnosis were collected during several sampling surveys between spring 2020 and autumn 2022 in several places across the Spanish Mediterranean coastline.

These samples were collected in putative sanctuaries, open sea and two different aquaria. A total of 26 samples of eDNA from open sea water were collected from several places: (i) Balearic Islands: Cabrera National Park (4), Ibiza (1), and Mallorca (1); (ii) Catalonia: Girona (3); (iii) Valencian Community: Alicante (4) and Columbretes Marine Reserve (5); (iv) Murcia Region: Cabo de Palos Marine Reserve (5) and Isla Grosa (3) (Figure 1, Table 1). Regarding putative sanctuaries, 3 samples from Ebro Delta (in 2020 and 2022) and 8 samples from Mar Menor Lagoon (from 2020 to 2022) were also collected. Additionally, 3 aquarium experiments were run with infected individuals of *P. nobilis* in 2021 and 2022, with 1 in Croatia (Pula) and 2 in Spain (IMEDMAR-UCV, in Calpe and IRTA, in Sant Carles de la Rápita). For this purpose, infected individuals were independently placed for 24 h in support containers into small aquaria with seawater in a closed circuit, without food. After 24 h, the water in these installations (~250 mL) was filtered and a total of 19 samples were collected using a vacuum pump and 0.45 μm membrane filters with 47 mm diameter.

Three methods were used for the collection of the eDNA seawater samples. (i) Water collection: 7 L to 20 L were placed into bottles at 1 m depth and, once in the lab, filtered through 0.45 µm cellulose acetate membranes using a vacuum pump. (ii) Planktonic nets: The net (mesh size of 80 µm, except for Calpe, which was 150 µm) was dragged for 5 to 15 min depending on plankton density. (iii) Membrane filters of 0.45 μm were placed in situ, attached to a buoy for several days in the Port of Calpe (Spain).

To prevent cross-contamination, all equipment used in the water collection and filtration steps, including plastic bottles, filter funnels and tweezers, was decontaminated using >0.1% sodium hypochlorite solution. For all samples, filter membranes were kept in Falcon tubes and preserved with different methods: 100% ethanol at room temperature (2020), in RNAlater or freezing at −20 °C (2021–2022), mainly depending on the different years of collection and possible maintenance for further DNA extractions.

### 2.2. Characterization of the H. pinnae Ribosomal Unit

For the characterization of the ribosomal unit of the parasite *H. pinnae*, mantle samples of infected *P. nobilis* were analyzed through a molecular approach. Total genomic DNA was extracted using the DNA NucleoSpin^®^ Tissue extraction kit (Macherey-Nagel, Duren, Germany) following the manufacturer’s instructions.

PCR and sequencing strategies using different primers were used to obtain the nucleotide sequence of the *H. pinnae* ribosomal unit region (Table 2). The quality and concentration of the DNA were measured using the Nanodrop ND1000 (Thermo Scientific; Waltham, MA, USA).

PCR reactions were performed in a total volume of 20 µL containing 10 µL KAPA Taq Ready Mix PCR kit (Sigma-Aldrich, Burlington, MA, USA), 0.4 µL of each primer (stock 20 µM) and 1 µL of DNA (20 up to 80 ng/µL) and water to make up the final volume. The temperature profile followed an initial denaturation at 94 °C for 2 min and 40 cycles of 94 °C for 30 s, 52–60 °C for 20 s and 72 °C for 1.45 min. PCR products were separated on 1.5% agarose in TAE 1x buffer gels (*w*/*v*), stained with GelRed (Biotium, Fremont, CA, USA) including a HighRanger 1 kb DNA ladder size standard (Norgen, Thorold, Canada), and visualized on a UV transilluminator. Obtained amplifications were cut and purified from agarose gel to ensure the specificity of amplification using the Metabion International mi-Gel Extraction Kit (Metabion International, Planegg, Germany) following the manufacturer’s instructions.

All obtained PCR fragments were bi-directionally sequenced using the 3130xl DNA automated sequencer (Applied Biosystems, Carlsbad, CA, USA) at Secugen S.L. service (Madrid, Spain).

The sequences were edited and aligned using the BioEdit v7.2.5 software [44] and MEGA X [45].

**Table 2 microorganisms-11-01146-t002:** List of primers used for ribosomal unit amplification and sequencing. Sequences are written in a 5′-to-3′ direction. Melting temperature (TM) of each primer and references are indicated.

Primer Name	5′–3′	TM °C	Reference
Forward			
HPNF1	AGCTTGACGGTAGGATATGGG	61	Catanese et al., 2018 [1]
18EUK581	GTGCCAGCAGCCGCG	57	Carnegie et al., 2003 [46]
HpF3	GCGGGCTTAGTTCAGGGG	61	López-Sanmartín et al., 2019 [27]
HapF1	GTTCTTTCWTGATTCTATGMA	53	Renault et al., 2017 [47]
HPNF3	CATTAGCATGGAATAATAAAACACGAC	62	Catanese et al., 2018 [1]
HPN18SF	CGCCTAGAAGCTCTGTGAACCTT	65	This study
HPNITSF	ACTGCGATAAGACTTGCGAACCGTCATTGTG	72	This study
LSU5	AGGTCGACCCGCTGAAYTTAAGCA	66	Olson et al., 2003 [48]
HPNITSsec2F	TCTTGAAACACGGACCAAGGAGTCT	66	This study
HPNITSsec1F	CACTTGGCAGTTTGATCCCGTAAAGC	68	This study
HPN28SF	TCTTAAGGTAGCCAAATGCCTCGTCA	66	This study
HPNIGSsec1	CCGTCGTGAGACAGGTTAGTTTTACCC	70	This study
Reverse			
HPNIGSR	CTCAGTTCTGCCTACTTCGGTCGT	67	This study
HPN18SR	GCTGCTAACCGTTATTTCTTGTCACTACCTC	71	This study
HpR3	AAAACCAACAAAGGCCCGAA	61	López-Sanmartín et al., 2019 [27]
18EUK1134	TTTAAGTTTCAGCCTTGCG	53	Carnegie et al., 2003 [46]
HPNR3	GCGACGGCTATTTAGATGGCTGA	65	Catanese et al., 2018 [1]
HapR2	GATGAAYAATTGCAATCAYCT	54	Renault et al., 2017 [47]
DiplostR4	TATGCTTAAATTCAGCGGGT	54	Galazzo et al., 2002 [49]
HPNITS2R	TACCACCTGCTTCATGCTACAATGTCGT	69	This study
1500R	GCTATCCTGAGGGAAACTTCG	61	Olson et al., 2003 [48]
HPN28ITSR	CCACGCCCGGCTGTCTCTATAAACTGA	71	This study

The primers HPNITSF and 1500R were then used to amplify a partial rDNA region (ITS2 and putative D1–D3 domains of 28S-rDNA) from tissues of *P. nobilis* sampled in different years and different areas of the Mediterranean Sea to evaluate the nucleotide variability and validate its efficiency for the possible PCR application to filtered water samples. The PCR products were cloned using a TOPO TA Cloning Kit (Invitrogen, Thermo Fisher, Waltham, MA, USA) according to the manufacturer’s protocol. Clones were sequenced using T3 and T7 universal primers, purified and bi-directionally sequenced. Sequencing of both strands was performed with 3 clones per rDNA fragment per sample.

### 2.3. Filtered Water Analyses

The DNA extraction was carried out using the DNeasy PowerWater Kit (Qiagen, Hilden, Germania), following the manufacturer’s instructions. The quality and concentration of the DNA were measured using the Nanodrop ND1000 (Thermo Scientific; Waltham, MA, USA).

Three fragments of the ribosomal unit were amplified by PCR using different primer pairs. For the largest-sized amplicon, the primer HPNITSF, designed from the sequences obtained in this paper (see above) and located between the putative 5.8S rDNA and the ITS region, was paired with the primer 1500R to amplify a DNA region that includes the ITS and a partial 28S rDNA gene (Figure 2, Table 2). The primers HapF1/HapR2 and HpF3/HpR3 were used to amplify partial regions of the 18S rDNA gene (Figure 2, Table 2). The PCR reactions were carried out as described above in a Biometra PCR Thermocycler (Göttingen, Germany) and consisted of a pre-denaturation period for 2 min at 94 °C and 40 cycles of 94 °C for 30 s, 49–60 °C for 20 s and 72 °C for 30 s–1.20 min. Annealing temperature and extension time varied in relation to the primer pairs used and the expected size of amplification.

### 2.4. Ethical Approval

Appropriate ethics, permissions to manipulate animals and other approvals were obtained for the submitted research. All biological samples were collected with the permission of regional and national authorities within the European Project LIFE Pinnarca, during management operations as part of the study plan approved by the Generalitat de Catalunya (Departament d’Acció Climàtica, Alimentació i Agenda Rural, Direcció General de Polítiques Ambientals i Medi Natural, Parc Natural del Delta de l’Ebre) and the Croatian Government (Ministry of Economy and Sustainable Development).

## 3. Results

### 3.1. Characterization of the Nuclear Ribosomal DNA Unit

The DNA sequence of the entire rRNA transcription unit was determined at LIMIA-IRFAP (Marine Research and Aquaculture Lab of the Balearic Islands Government). Obtained fragments were aligned and assembled to generate a single edited sequence. It was 5978 bp in length, including the small subunit (SSU 18S) rDNA, internal transcribed spacer 1 (ITS1), 5.8S rDNA, internal transcribed spacer 2 (ITS2), large subunit (LSU 28S) rDNA and partial intergenic spacer (IGS). The whole sequence was submitted to the GenBank/EMBL/DDBJ database, where it is available under the accession number LC637522. The base composition in the dataset was balanced with a mean G+C content of 55.42% for the aligned complete sequences.

Alignments with 18S–ITS1–5.8S–ITS2–28S partial sequences of *Haplosporidium costale* from *Crassostrea virginica* (KF790901–2 [50]) and from *C. gigas* (MZ666374 [51]), of *Haplosporidium* sp. from *Saccostrea glomerata* (KF790894–900 [50]), and of *Haplosporidium littoralis* (KJ150289 [52]) were used to compare and to allocate the start and stop nucleotide locations of rRNAs. Despite a challenging evaluation due to extensive homology with known sequences, we propose a putative length size of 1910 bp and 2969 bp for consensus sequences of 18S rDNA and LSU 28S rDNA. The size of the putative 5.8S rDNA gene was 143 bp. In addition, comparing the 5.8S sequence of *H. pinnae* with those from GenBank, *H. littoralis* (accession no. KJ150289), *Bonamia exitiosa* (accession no. KY680577-KY680645), *Cryptosporidium parvum* (accession no. MZ892386-MZ892388) and *Perkinsus atlanticus* (accession no. AF509333) conserved regions were detected (5′-ATGGATGHCTHGGYTC-3′, 5′-CGAKGAAGRACGC-3′ and 5′-YGCGATAV-3′). Finally, the putative lengths of ITS1 and ITS2 sequences for *H. pinnae* were 128 bp and 196 bp, respectively.

Moreover, sequences of cloned PCR products of a partial region including ITS2 and 28S rDNA of *H. pinnae* were obtained from the mantle of adult *P. nobilis.* The sequence of the obtained amplicons (983 bp length) did not show variations in nucleotide composition among the samples collected in different areas of the Mediterranean Sea. Only one haplotype was observed among all the studied sequences, revealing the absence of polymorphism.

### 3.2. H. pinnae Detection from Water Samples

Fragments of about 980 bp (fragment A), 350 bp (fragment B) and 165 bp (fragment C) were obtained from the three different PCR primer pairs employed. To ensure that the amplicons were specific to *H. pinnae*, some of the amplified fragments were cut from the agarose gel, purified and sequenced. The resulting sequences showed 100% identity with the consensus sequence obtained from *H. pinnae* isolated in adult *P. nobilis* individuals (accession number LC637522).

The results of the analyzed filters showed a percentage of positive amplification of *H. pinnae* DNA varying from 9.1% to 53.6%, considering all the conservation methods and amplified fragments (Table 3). The lowest values were observed using ethanol conservation while the highest corresponded to frozen filters. However, considering all conservation methods, differences in amplification from the longest to the shortest fragment were observed, indicating an evident degradation of *H. pinnae* DNA in water, regardless of the type of storage used. In fact, the amplification varied from 0% in the longest fragment (fragment A) to 9.1% in the shortest fragments (fragments B and C) for samples preserved in ethanol, from 17.6% (fragments A and B) to 29.4% (fragment C) for samples stored in RNAlater and from 25% (fragment A) to 53.6% (fragment C) for frozen samples (Table 3).

In regard to the detection of *H. pinnae* by PCR amplification in water samples from the different environments, the results showed a similar pattern, albeit with different percentage values.

Positive amplifications from 85.7% to 100% were obtained for aquarium water samples, from 28.6% (fragment A) to 57.1% (fragment C) for filters stored in RNAlater (total 87.5%) and from 58.3% (fragment A) to 100% (fragment C) for frozen filters. Samples from this environment were not preserved in ethanol (Table 3).

In relation to water samples collected from the open sea, positive detections were 14.3%, 12.5% and 27.3% for those preserved in ethanol, RNAlater and frozen, respectively. In ethanol, they were 14.3% (fragments B and C); in RNAlater, 12.5% (fragments A and C); and in frozen filters, 27.3% (fragment C) (Table 3).

Finally, for water samples collected in putative sanctuaries, the positive values were 0% for those preserved in ethanol, RNAlater and frozen (Table 3).

As expected, regardless of the fragment used, the samples of aquaria showed the highest results for parasite detection. On the contrary, no positive samples were detected in putative sanctuaries. In the open sea, the percentage of positive samples varied from 14.3% in 2020 to 25% in 2021, and then dropped to 18% in 2022 (Figure 3).

## 4. Discussion

In this study, the complete rDNA unit of *H. pinnae* was sequenced for the first time. Moreover, our results provide the first steps of a method that allows the detection of *H. pinnae* in water samples.

As already described in other studies, this parasite seems to be molecularly and morphologically different from other species of the same order [1,18]. Nevertheless, the putative *H. pinnae* SSU 18S rDNA gene (1910 bp in length) resulted as longer than that of *H. littoralis* (1736 bp [52]) and *H. costale* (1791 bp [53]) and *Bonamia* spp. (~1750–1766 bp [54]), whereas it was similar to *Haplosporidium montforti* (1872 bp [55]) and in the range of some species of marine cestodes (1841–1980 bp [56]). Regarding the *H. pinnae* 28S rDNA gene (2969 bp), the limited availability of complete LSU rDNA sequences from Haplosporidan species in GenBank makes its characterization more difficult. The only described *Haplosporidium* 28S rDNA in GenBank is *H. littoralis*, with a length of 3031 bp [52].

The ribosomal genes form a cistron that behaves essentially as a single locus, despite the presence of multiple tandem copies in the genome. In fact, copies of the ribosomal unit within the same genome do not evolve independently but follow what is called a concerted evolution: the members of a multigenic family are rapidly homogenized by crossing over unequal or genetic conversion [57]. However, the lack of nucleotide variability in ITS2 and 28S rDNA observed among all the analyzed fragments of *H. pinnae* in this study makes us consider this DNA region suitable for species identification, due to sequence conservation. Although eukaryotic 28S rRNA normally consists of twelve divergent domains (D1–D12), which alternate with a more or less conserved sequence [58], it seems evident from obtained sequences for *H. pinnae* that the first domains in this gene exhibit a low rate of divergence.

It is also possible that we did not observe some rare variants in this study, due to the limited number of clones sequenced. However, such variations, if they occur at a very low frequency, would not affect the quality of direct sequencing and its usefulness in species identification or phylogenetic studies. In fact, the region ITS2-28S rDNA is considered a nuclear marker easily to amplify. In a similar study of calanoid copepods, it presented low variability within species (it did not need to be cloned) but was sufficiently variable for being used as a possible marker for DNA barcoding or to identify genetic groups [59].

In our work, samples collected from controlled aquarium conditions both in Spain and Croatia showed the highest number of positive *H. pinnae* detections. This is probably due to a higher concentration of the pathogen compared to the open sea despite a lower amount of filtered seawater.

On the other hand, in this study, no positive parasite detections were found in any of the sanctuaries, unlike some authors who detected the parasite in both feces and/or mantle tissue of individuals living in the Mar Menor Lagoon [60,61] and in Delta Ebro [62]. The lack of positive *H. pinnae* detections in the sanctuaries could be due to the absence of the parasite as a consequence of the protective barrier generated by some chemical–physical features typical of paralic environments, to the low concentration of the parasite in that sampling site (collected away of infected *P. nobilis*) or to the low number of samples analyzed.

Previous studies have described more individuals of *P. nobilis* infected with *H. pinnae* in the open sea than in coastal harbors and lagoons, probably due to physical or environmental barriers (i.e., salinity) [18,62,63]. In fact, the spreading of *H. pinnae* has led to the existence of a few areas with relict *P. nobilis* populations, located in coastal lagoons and river deltas [62,63,64]. In general, analytical detection in these environments can be more complicated because there may be a low concentration of the parasite, but it may be present with inherent risk to populations. For this reason, we recommend intensive and periodic sampling in order to ensure that the coastal lagoons are truly free of the parasite. Cases of *H. pinnae* detection within some lagoons (sometimes referred to as sanctuaries) are also being observed and have been described as a consequence of parasite spread from open waters, resulting in enhanced mortalities in the outer region closer to the mouth of the lagoons or in the seaward vicinity of connecting channels [61,64,65,66,67]. In support of this, in 2018, the population in Alfacs Bay (South Ebro Delta) was infected only in its more external part located next to the open sea and subjected to higher salinities [68]. Anyway, although the spread of pathogens was associated with cumulative mortality in Alfacs Bay, it seems to have been hindered by a salinity gradient of 37.4–35.7% (100% mortality near the mouth, 43% in middle regions and 13% in inner regions) [62].

However, although it was possible to associate the spread of the parasite in the western Mediterranean Sea to marine currents [63], the presence and life cycle stage of *H. pinnae* in the water column have been unreported up to now. Our results for the analyses of seawater samples from the open sea support this topic. In fact, although with low abundance, the presence of *H. pinnae* was detected in samples from different areas of the Spanish Mediterranean coast. In this context, it is important to remark on the existence of a positive signal of *H. pinnae* in areas where the entire population of *P. nobilis* has been dead since 2016 [1,5]. Positive values of the existence of the pathogen in these areas after more than 5 years point to the probable existence of intermediate hosts that are playing an important role in the preservation of the pathogen in the environment.

Parasites with direct life cycles spend most of their adult life in one host in multiple developmental stages, with their offspring being passed on from one host to another. For the free-living stage, they must be able to survive in an environment outside their original host (cells or spores) and then locate and settle into a new host. In contrast, parasites with indirect life cycles are characterized by two host stages, requiring a definitive host and an intermediate host. The definitive host stage is required for reproduction and the adult life stage, but parasite development occurs within the intermediate host, after which it can be transmitted to a definitive host. The occurrence of different life cycle stages of *H. pinnae* in the same individual, as observed in other studies, is not common among haplosporidan species and suggests a direct life cycle [1,18]. However, the dynamics of haplosporidans in their hosts, annual cycle and infection periods are seasonal and depend on environmental conditions [69]. Spore stages and sporulation processes have never been detected in *B. ostreae*, *B. exitosa* [3] and *H. littoralis* [70]. Likewise, neither uni- nor binucleated stages are common in other species of the genus Haplosporidium such as *H. tapetis* [71], *H. nelsoni* [72] or *H. edule* [73]. Thus, considering the different life stages of *H. pinnae*, it could be possible to find either spores or free uni-bi nucleate cells as infective forms of the parasite in seawater.

Nevertheless, what we can deduce from our results is that degradation in water probably has an important role in the spreading and dynamic mechanisms of the parasite. PCR amplifications of larger fragments, as clearly shown by our data, were lower than for smaller ones. Water-filtered samples, in which living juveniles of *P. nobilis* have been housed, had been already used for developing a rapid PCR procedure for discrimination between the two Mediterranean *Pinna* species [43]. However, the application of this kind of eDNA method shows the inconvenience of DNA degradation after a short time, so that only small fragments can be detected. Long DNA fragments have a faster degradation rate than short DNA fragments [74], resulting in lower sensibility for detection analysis. Therefore, the application of three PCRs generating amplicons of different lengths, as shown in this work, could offer the possibility to observe different levels of *H. pinnae* DNA degradation, favoring the understanding and identification of the moment at which the parasite has the ability to infect or already features a disrupted activity.

Based on the results obtained, we consider that this study achieved the development of an efficient methodology to detect the presence of the parasite *H. pinnae* in water samples from different environments: aquaria, open sea, and sanctuaries. The results also showed the ability to detect *H. pinnae* in seawater from different areas, but with different degrees of DNA fragmentation, analyzing them with three PCRs generating amplicons of different lengths.

For future studies aimed to detect *H. pinnae* in seawater, we strongly recommend freezing the filters since it was observed to be the best method for storing them, rather than preserving them in ethanol or RNAlater. The use of the three PCRs generating amplicons of different lengths is also recommended as it provides better knowledge of the degradation status of the parasite. The amplicons of different lengths could even recognize the level of infection of the juveniles *P. nobilis* retained in the collectors installed throughout the Mediterranean Sea. Moreover, since the rate of degradation and mobilization of DNA in seawater is still unknown, we suggest carrying out more detailed studies, collecting and analyzing more water samples, particularly in areas here defined as possible sanctuaries, to ensure the effective absence of the parasite.

In conclusion, this method offers a valuable tool for better understanding the life cycle of *H. pinnae* and its spread, and consequently, for conducting preventive seawater analyses in areas where there is interest in identifying parasite-free zones for implanting juvenile *P. nobilis* from larvae collectors or for future mollusk culture.

## Figures and Tables

**Figure 1 microorganisms-11-01146-f001:**
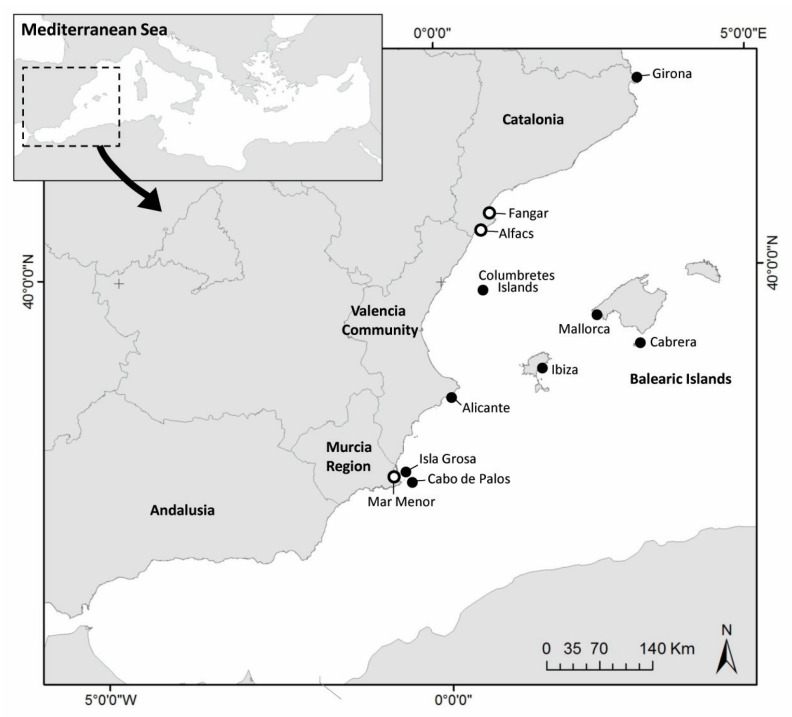
Map of sampling sites in Western Mediterranean Sea. Open sea (black circles) and putative sanctuary areas (white circles).

**Figure 2 microorganisms-11-01146-f002:**
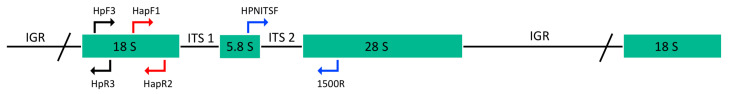
Schematic representation of ribosomal RNA gene unit and PCR assays used in this study. The position of forward primers paired with reverse primers, for PCR amplifications of 980 bp (blue), 350 bp (red) and 165 bp (black) fragments are indicated in different colors.

**Figure 3 microorganisms-11-01146-f003:**
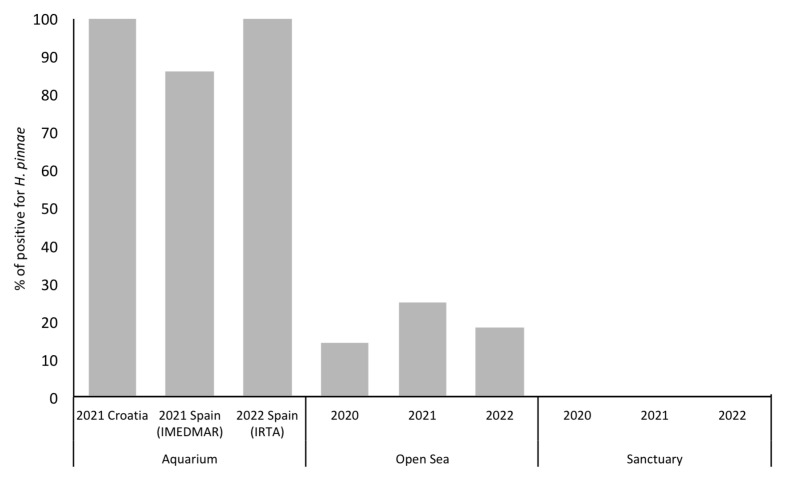
Percentage of *H. pinnae* positive detection in water samples.

**Table 1 microorganisms-11-01146-t001:** List of sites and the number of samples collected for molecular analyses. P: plankton, W: water, Wf: membrane filters in situ.

			2020	2021	2022
Sanctuary	Delta Ebro	Alfacs	1W		1P
		Fangar			1P
	Mar Menor	Baron	1W	1P	1P
		Pueblo Calido	1W	1P	
		Perdiguera	1W		1P
		Pedrucho			1P
Open Sea	Balearic Islands	Cabrera	3P		1P
		Mallorca	1P		
		Ibiza			1P
	Murcia Region	Isla Grossa	1P	2P	
		Cabo de Palos	2P	2P	1P
	Valencian Community	Alicante		1W, 3Wf	
		Columbretes			5P
	Catalonia	Girona			3P

**Table 3 microorganisms-11-01146-t003:** Results of the three PCRs analyses of water, based on environment type and storing method of the samples. Positive amplifications in relation to analyzed samples and percentages are shown. All fragments are the total number of positive individuals detected, regardless of the different PCR fragments.

		All Fragments	Fragment A	Fragment B	Fragment C
All samples	Ethanol	1/11 (9.1%)	0/11 (0%)	1/11 (9.1%)	1/11 (9.1%)
	RNAlater	7/17 (41.2%)	3/17 (17.6%)	3/17 (17.6%)	5/17 (29.4%)
	Frozen Filter	15/28 (53.6%)	7/28 (25%)	10/28 (35.7%)	15/28 (53.6%)
Aquarium	Ethanol	-	-	-	-
	RNAlater	6/7 (85.7%)	2/7 (28.6%)	3/7 (42.9%)	4/7 (57.1%)
	Frozen Filter	12/12 (100%)	7/12 (58.3%)	10/12 (83.3%)	12/12 (100%)
Open sea	Ethanol	1/7 (14.3%)	0/7 (0%)	1/7 (14.3%)	1/7 (14.3%)
	RNAlater	1/8 (12.5%)	1/8 (12.5%)	0/8 (0%)	1/8 (12.5%)
	Frozen Filter	3/11 (27.3%)	0/11 (0%)	0/11 (0%)	3/11 (27.3%)
Sanctuary	Ethanol	0/4 (0%)	0/4 (0%)	0/4 (0%)	0/4 (0%)
	RNAlater	0/2 (0%)	0/2 (0%)	0/2 (0%)	0/2 (0%)
	Frozen Filter	0/5 (0%)	0/5 (0%)	0/5 (0%)	0/5 (0%)

## Data Availability

The data that support the findings of this study are available in GenBank via accession number LC637522.

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
