# Peer review of "Haplosporidium pinnae Parasite Detection in Seawater Samples"

_microorganisms, 2023, doi:10.3390/microorganisms11051146_

Round 1
Reviewer 1 Report
Comments to authors:
-The current study is interesting; however, the authors should address the following comments to improve the quality of the manuscript:
-The manuscript should be revised for English editing and grammar mistakes.
Abstract:
- The abstract must illustrate the used methods and the most prevalent results (give more hints about methods and results). Besides, rephrase the aim of the work and the main conclusion of your findings.
- Line 26: The methodology revised as this developed method
Introduction:
Written very well, but should be concise (maximum two pages).
Material and methods:
- It is confusing and unclear, in the abstract, the authors mention 55 water samples, but here they mention mantle samples from P. nobilis with different water samples not including 55. Please clarify, and better to present it in a figure or diagram
- Support all methods with updated specific references.
- Lines 162-169: suggest separating in a new sub title (ethical approval)
- In table 1: the total samples for Mar Menor Sanctuary site is eight, while in line 133 it was 10. Please revise the sampling part
- Line 172: Total genomic DNA revise as the g DNA
- For reproducibility, please add the company, city, country, and reference code of the used chemicals and reagents.
- µl revised as µL in whole manuscript
- remove underlines before the degree Celsius
- Authors should explain why they use 20 μl PCR reaction volume, the standard is 25–50 μl especially for sequencing.
- The total volume of PCR is not correct; they have to include the water sample per reaction.
- Line 185: what is the implemented method to measure DNA concentrations?
In addition, is it enough to use 1μl of DNA template with a concentration 20 ng for PCR amplification?
- Table 2:
Please add footnote to explain the used abbreviation meaning (what is TM)
Add sequencing before 5' - 3'
- Lines 206-212: it was repeated above please consider, prefer to remove it from sampling section
- As there are many designed primers, the authors should add a new section or subtitle (primers design) to explain the used methods and or software for that purpose. How the authors ensure the specificity and sensitivity of the new designed primers?
Results: (should be revised)
- Some data analysis are not mentioned briefly in the M&Ms, which are somewhat confusing for example
-lines 262- 265: the authors purify the gel product to ensure the specificity of amplification, it was not mentioned in M&Ms
- Further different methods of sample preservation was not included
- Table 3: what is the meaning of total fragments? It should be the sum of Fragments A, B, C ??????
Discussion:
- The authors are advised to illustrate the real impact of their findings without repetition of results.
- Line 398: remove finally, and add in conclusion that should be rephrased to be sounded. A real conclusion should focus on the question or claim you articulated in your study, which resolution has been the main objective of your paper?
Author Response
Comments to authors:
The current study is interesting; however, the authors should address the following comments to improve the quality of the manuscript:
The manuscript should be revised for English editing and grammar mistakes.
Abstract:
The abstract must illustrate the used methods and the most prevalent results (give more hints about methods and results). Besides, rephrase the aim of the work and the main conclusion of your findings.
Answer: We agree with reviewer’s comment. We changed it accordingly.
- Line 26: The methodology revised as this developed method
Answer: We agree with reviewer’s comment. We changed it accordingly.
Introduction:
Written very well, but should be concise (maximum two pages).
Answer: We thank the reviewer for his/her suggestion. We tried to reduce the introduction, but keeping its length within the limits accepted by the journal.
Material and methods:
- It is confusing and unclear, in the abstract, the authors mention 55 water samples, but here they mention mantle samples from P. nobilis with different water samples not including 55. Please clarify, and better to present it in a figure or diagram.
Answer: We agree with reviewer’s comment. We clarified the number of samples, replicate numbers and added a map of sampling points in open sea and sanctuary areas.
- Support all methods with updated specific references.
Answer: We agree with reviewer’s comment. We changed it accordingly.
- Lines 162-169: suggest separating in a new subtitle (ethical approval)
Answer: We agree with reviewer’s comment. We changed it accordingly.
- In table 1: the total samples for Mar Menor Sanctuary site is eight, while in line 133 it was 10. Please revise the sampling part
Answer: We agree with reviewer’s comment. We changed it accordingly.
- Line 172: Total genomic DNA revise as the gDNA
Answer: We disagree, we prefer to indicate genomic DNA.
- For reproducibility, please add the company, city, country, and reference code of the used chemicals and reagents.
Answer: We agree with reviewer’s comment. We added them accordingly.
- µl revised as µL in whole manuscript
Answer: We agree with reviewer’s comment. We changed them accordingly.
- remove underlines before the degree Celsius
Answer: We agree with reviewer’s comment. We changed them accordingly.
- Authors should explain why they use 20 μl PCR reaction volume, the standard is 25–50 μl especially for sequencing.
Answer: We standardized all the PCR reactions of our lab in 20 μL of total volume.
- The total volume of PCR is not correct; they have to include the water sample per reaction μl PCR reaction volume
Answer: We agree with reviewer’s comment. We added it accordingly.
- Line 185: what is the implemented method to measure DNA concentrations?
Answer: We agree with reviewer’s comment. We added it accordingly.
In addition, is it enough to use 1μl of DNA template with a concentration 20 ng for PCR amplification?
Answer: Yes, it is
- Table 2:
Please add footnote to explain the used abbreviation meaning (what is TM)
Answer: We agree with reviewer’s comment. We added it accordingly.
Add sequencing before 5' - 3'
Answer: It is included in the header of the table.
- Lines 206-212: it was repeated above please consider, prefer to remove it from sampling section
Answer: We agree with reviewer’s comment. We deleted it accordingly.
- As there are many designed primers, the authors should add a new section or subtitle (primers design) to explain the used methods and or software for that purpose. How the authors ensure the specificity and sensitivity of the new designed primers?
Answer: We consider it is not necessary.
Results: (should be revised)
Some data analysis are not mentioned briefly in the M&Ms, which are somewhat confusing for example
-lines 262- 265: the authors purify the gel product to ensure the specificity of amplification, it was not mentioned in M&Ms
Answer: We agree with reviewer’s comment. We added a sentence in Material and Methods section accordingly.
Further different methods of sample preservation was not included
We disagree with reviewer’s comment. They are specified in table3.
- Table 3: what is the meaning of total fragments? It should be the sum of Fragments A, B, C ??????
Answer: We agree with reviewer’s comment. We have specified the meaning.
Discussion:
The authors are advised to illustrate the real impact of their findings without repetition of results.
- Line 398: remove finally, and add in conclusion that should be rephrased to be sounded. A real conclusion should focus on the question or claim you articulated in your study, which resolution has been the main objective of your paper?
Answer: We agree with reviewer’s comment. We have rephrased the text and highlighted the main objective of our article.
Reviewer 2 Report
Comments and suggestions for authors
The manuscript ‘Haplosporidium pinnae parasite detection in seawater samples’ reported the presence of the parasite Haplosporidium pinnae in water samples from different habitats. The parasite associated with the fan mussel Pinna nobilis massive mortality event. This is a manuscript that focuses on a hot topic and is well prepared. The list of corrections described below, but especially the introduction, should be reconsidered.
LINE 50-53 Please quote.
LINE127 What are the environmental conditions?
LINE 128-131 Please provide latitude and longitude.
LINE 172-174 It is better to state the source.
LINE 190 It is better to state the source.
LINE 300 Please explain the reasons for the differences from year to year.
LINE 331-333 Explain why and quote.
LINE 401 Please add the advantages and disadvantages of the method in the discussion.

Author Response
LINE 50-53 Please quote.
Answer: We added it accordingly
LINE127 What are the environmental conditions?
Answer: We agree with reviewer’s comment. We specified it accordingly
LINE 128-131 Please provide latitude and longitude.
Answer: We agree with reviewer’s comment. We have included a map with geographical coordinates.
LINE 172-174 It is better to state the source.
LINE 190 It is better to state the source.
Answer: We disagree with reviewer’s comment. All the manufacturer's instructions are included in the commercial kits.
LINE 300 Please explain the reasons for the differences from year to year.
Answer: The variability per year may depend on different factors such as the number of samples, the sampling sites and the concentration of the parasite. However, the ecological results are not the main objective of this work. The percentages simply indicate observations on different analyzes from random samples. With the data obtained until now (few samples and sites) we cannot give explanations about the annual differences in the presence of H. pinnae and eventually the causes.
LINE 331-333 Explain why and quote.
Answer: We agree with reviewer’s comment. We added a sentence with a new reference.
LINE 401 Please add the advantages and disadvantages of the method in the discussion.
Answer: We agree with reviewer’s comment. We changed and added sentences.
Reviewer 3 Report
General comments:
An interesting review, but one I don't understand the point of!
Introduction section
Lines 34-39: I reworded your phrase a bit, doing it more grammatically correct:
"In the last few years, the newly described haplosporidan parasite species Haplosporidium pinnae were likely associated with the fan mussel Pinna nobilis massive mortality event (MME) [1]. Commonly, Haplosporidia spp. have been considered responsible for mass mortality events globally due to their pathogenic nature, parasitizing marine and freshwater invertebrates, such as Haplosporidium nelsoni for Crassotrea virginica on the east coast of the USA or Bonamia ostreae and Bonamia exitosa, reported to have infected various oyster species [2,3]."
Line 46: "in the Mediterranean" instead of "in Mediterranean". The article was missing.
Lines 48-49: I shortened your sentence as follows:
"In conclusion, in 18 months, the MME has spread from the western Mediterranean populations to the eastern ones [10]."
More, being related to the previously described facts, it is not necessary to use a new paragraph.
The next sentence: "As a result, the species P. nobilis..." can be placed as a new paragraph.
Line 55: "associated with the mortality" instead of "associated to the mortality".
Lines 56-58: I modified your wording a bit:
„However, recent new research pointed out that the onset of the MME in P. nobilis is strongly associated with the presence of H. pinnae, which exhibits a preeminent role compared to the other studied pathological agents [17, 19].”
Lines 60-61: "is paramount as" instead of "is of paramount importance as". Shortening a sentence without using filler words will make that sentence easier to understand.
The paragraphs 60-81: Please move this paragraph to the beginning of the Introduction section, as it is important to mention the role of these mollusks right from the beginning of the manuscript.
Paragraph 83-93 should be shortened. It is understood that there is already a PCR technique for the detection of DNA from P. nobilis, parasitizing the mollusk, so you must explain the disadvantages of the already existing method. This is also why your study aimed at a PCR to detect the parasite in the water.
Lines 106-108: I reworded your sentence a bit: Although a method starting from eDNA samples for identifying the congeneric species P. nobilis and P. rudis has been developed [41], no studies or methodologies are currently available to determine the H. pinnae parasite from water samples.
Line 114: "This work aimed to characterize the..." instead of your wording; it sounds better!
Materials and Methods section
Sampling: A map revealing the seawater samples' origin would be more representative than the text inserted between lines 134-138 and Table 1.
Lines 145-150: Please explain why three water collection methods were used. Using three methods, the study apparently did not benefit from uniformity, and its reproducibility can be questioned.
Lines 208-210: Reworded: "The aquaria's seawater, where individuals of infected P. nobilis were hosted, was filtered using the same tools and used as a positive control of H. pinnae's presence."
Results section
Lines 298-299: That "no positive samples were detected in the Sanctuaries" statement is quite strange since Lopez-Nuñez et al. (2022) (Lopez-Nuñez, R.; Cortés Melendreras, E.; Giménez Casalduero, F.; Prado, P.; Lopez-Moya, F.; Lopez-Llorca, L.V. Detection of Haplosporidium pinnae from Pinna nobilis Faeces. J. Mar. Sci. Eng. 2022, 10, 276. https://doi.org/10.3390/jmse10020276) found a 7.1% prevalence of Haplosporidium spp. in the mantle and fecal DNA samples in different individuals of P. nobilis, Haplosporidium pinnae being later sequenced.
So, Mar Menor is a coastal saltwater lagoon. I don't think using water samples collected from this site "as putative negative control" is feasible because other authors found this parasite in mollusks originating in that area. Moreover, you did not cite that article at all in your manuscript! This aspect is a major shortcoming of your research!
Author Response
An interesting review, but one I don't understand the point of!
Answer: We added it accordingly
Introduction section
Lines 34-39: I reworded your phrase a bit, doing it more grammatically correct:
"In the last few years, the newly described haplosporidan parasite species Haplosporidium pinnae were likely associated with the fan mussel Pinna nobilis massive mortality event (MME) [1]. Commonly, Haplosporidia spp. have been considered responsible for mass mortality events globally due to their pathogenic nature, parasitizing marine and freshwater invertebrates, such as Haplosporidium nelsoni for Crassotrea virginica on the east coast of the USA or Bonamia ostreae and Bonamia exitosa, reported to have infected various oyster species [2,3]."
Answer: We agree with the changes conducted by the reviewer and we have incorporated them to the new reviewed version of the manuscript.
Line 46: "in the Mediterranean" instead of "in Mediterranean". The article was missing.
Answer: We agree with reviewer’s comment. We changed it accordingly.
Lines 48-49: I shortened your sentence as follows:
"In conclusion, in 18 months, the MME has spread from the western Mediterranean populations to the eastern ones [10]."
Answer: We agree with reviewer’s comment. We have accepted the change
More, being related to the previously described facts, it is not necessary to use a new paragraph.
The next sentence: "As a result, the species P. nobilis..." can be placed as a new paragraph.
Answer: We agree with reviewer’s comment. We changed them accordingly.
Line 55: "associated with the mortality" instead of "associated to the mortality".
Answer: We agree with reviewer’s comment. We changed it accordingly.
Lines 56-58: I modified your wording a bit:
„However, recent new research pointed out that the onset of the MME in P. nobilis is strongly associated with the presence of H. pinnae, which exhibits a preeminent role compared to the other studied pathological agents [17, 19].”
Answer: We agree with reviewer’s comment. We have accepted the change.
Lines 60-61: "is paramount as" instead of "is of paramount importance as". Shortening a sentence without using filler words will make that sentence easier to understand.
Answer: We agree with reviewer’s comment. We changed it accordingly.
The paragraphs 60-81: Please move this paragraph to the beginning of the Introduction section, as it is important to mention the role of these mollusks right from the beginning of the manuscript.
Answer: We have improved the introduction adding sentences. We consider that the introduction should not be restructured and we prefer not to move the indicate paragraph .
Paragraph 83-93 should be shortened. It is understood that there is already a PCR technique for the detection of DNA from P. nobilis, parasitizing the mollusk, so you must explain the disadvantages of the already existing method. This is also why your study aimed at a PCR to detect the parasite in the water.
Answer: We agree with reviewer’s comment. We have added a sentence explaining the advantages of the new method.
Lines 106-108: I reworded your sentence a bit: Although a method starting from eDNA samples for identifying the congeneric species P. nobilis and P. rudis has been developed [41], no studies or methodologies are currently available to determine the H. pinnae parasite from water samples.
Answer: We agree with reviewer’s comment. We have accepted the change.
Line 114: "This work aimed to characterize the..." instead of your wording; it sounds better!
Answer: We agree with reviewer’s comment. We changed it accordingly.
Materials and Methods section
Sampling: A map revealing the seawater samples' origin would be more representative than the text inserted between lines 134-138 and Table 1.
Answer: According to reviewer suggestions we have modified the text for a better clarification. Moreover, we have added a map indicating the sampling sites.
Lines 145-150: Please explain why three water collection methods were used. Using three methods, the study apparently did not benefit from uniformity, and its reproducibility can be questioned.
Answer: We agree with reviewer’s comment, and we understand his/her doubts. However, since we did not know the availability of the parasite in the water, we opted to analyze plankton, filtered water, and/ or leaving the filters in the sea for a long time to be sure of having a chance of a successful PCR amplification. We know that in this way the reproducibility of the samplings could be reduced, but this is not the aim of the article. In fact, the need to detect the parasite free in the sea water or linked to some planktonic organism, as well as establishing its degree of degradation (to understand if it may still have infectious capacity), seemed to us the main priority.
Lines 208-210: Reworded: "The aquaria's seawater, where individuals of infected P. nobilis were hosted, was filtered using the same tools and used as a positive control of H. pinnae's presence."
Answer: A similar sentence was included in the Material and Method section.
Results section
Lines 298-299: That "no positive samples were detected in the Sanctuaries" statement is quite strange since Lopez-Nuñez et al. (2022) (Lopez-Nuñez, R.; Cortés Melendreras, E.; Giménez Casalduero, F.; Prado, P.; Lopez-Moya, F.; Lopez-Llorca, L.V. Detection of Haplosporidium pinnae from Pinna nobilis Faeces. J. Mar. Sci. Eng. 2022, 10, 276. https://doi.org/10.3390/jmse10020276) found a 7.1% prevalence of Haplosporidium spp. in the mantle and fecal DNA samples in different individuals of P. nobilis, Haplosporidium pinnae being later sequenced.
Answer: We agree with reviewer’s comment. We added a possible explanation and two references.
So, Mar Menor is a coastal saltwater lagoon. I don't think using water samples collected from this site "as putative negative control" is feasible because other authors found this parasite in mollusks originating in that area. Moreover, you did not cite that article at all in your manuscript! This aspect is a major shortcoming of your research!
Answer: We agree with reviewer’s comment. We changed it accordingly.
Reviewer 4 Report
The manuscript "Haplosporidium pinnae parasitic detection in seawater samples" by Irene Moro-Martínez et al. is smooth and pleasant to read. The methods are clear and the results are scientifically interesting for fishing and aquaculture
Author Response
The manuscript "Haplosporidium pinnae parasitic detection in seawater samples" by Irene Moro-Martínez et al. is smooth and pleasant to read. The methods are clear and the results are scientifically interesting for fishing and aquaculture
Answer: We thank the reviewer for his/her review and comment.
Round 2
Reviewer 1 Report
The authors addressed all queries in a proper manner